# Management of Cancer-Related Cognitive Impairment: A Systematic Review of Computerized Cognitive Stimulation and Computerized Physical Activity

**DOI:** 10.3390/cancers13205161

**Published:** 2021-10-14

**Authors:** Giulia Binarelli, Florence Joly, Laure Tron, Sophie Lefevre Arbogast, Marie Lange

**Affiliations:** 1Clinical Research Department, Centre François Baclesse, 14000 Caen, France; giulia.binarelli@unicaen.fr (G.B.); f.joly@baclesse.unicancer.fr (F.J.); 2ANTICIPE, INSERM, UNICAEN, Normandie University, 14000 Caen, France; laure.tron@inserm.fr (L.T.); s.lefevre-arbogast@baclesse.unicancer.fr (S.L.A.); 3Cancer and Cognition Platform, Ligue Nationale Contre le Cancer, 14000 Caen, France

**Keywords:** cancer, cancer-related cognitive impairment, computerized cognitive stimulation, computerized physical activity, cognitive intervention

## Abstract

**Simple Summary:**

Many patients in oncology reports cognitive complaints with consequences on quality of life and seek support. Several interventions have been investigated to improve these symptoms, and, to date, cognitive stimulation and physical activity seem to be the most efficient. Nevertheless, these interventions are difficult to set up in a supportive care routine because they require the presence of professionals such as neuropsychologists and physical therapists, who are not always available. To overcome these barriers, computerized interventions have started to be investigated. This systematic review aimed to present the state of knowledge on computerized interventions based on cognitive stimulation and/or physical activity to improve cognitive difficulties in cancer patients. Both computerized physical activity and cognitive stimulation are efficient for cognitive improvement, although further investigation is necessary to compare efficiency between the two interventions and to investigate the possible added value of a combined intervention.

**Abstract:**

Cancer-related cognitive impairment (CRCI) occurs frequently in patients living with cancer, with consequences on quality of life. Recently, research on the management of these difficulties has focused on computerized cognitive stimulation and computerized physical activity programs. This systematic review presents the state of knowledge about interventions based on computerized-cognitive stimulation and/or physical activity to reduce CRCI. The review followed the PRISMA guidelines. A search was conducted in PUBMED and Web of Science databases. Risk of bias analysis was conducted using the Rob2 tool and the quality of evidence was conducted following the GRADE approach. A total of 3776 articles were initially identified and 20 of them met the inclusion criteria. Among them, sixteen investigated computerized-cognitive stimulation and four computerized-physical activity. Most of the studies were randomized controlled trials and assessed the efficacy of a home-based intervention on objective cognition in adults with cancer. Overall, cognitive improvement was found in 11/16 computerized-cognitive stimulation studies and 2/4 computerized-physical activity studies. Cognitive stimulation or physical activity improved especially cognitive complaints, memory, and attention. These results suggest the efficacy of both computerized-cognitive stimulation and physical activity. However, we report a high risk of bias for the majority of studies and a low level of quality of evidence. Therefore, further investigations are needed to confirm the efficacy of these interventions and to investigate the possible added benefit on cognition of a combined computerized-cognitive/physical intervention.

## 1. Introduction

Recently, the long-term adverse effects of cancer treatments and their impact on quality of life have been gaining attention, thanks to progress in medicine and the resulting gain in life expectancy of cancer patients. Cognitive difficulties after cancer treatments, also named cancer-related cognitive impairment (CRCI), are one of the most frequent side-effects reported by cancer patients (40–75% of them), mainly after chemotherapy (also named “chemobrain”) [1,2,3,4,5].

Cancer survivors mostly report memory decline, difficulty with focusing attention, word-finding, and a decrease in multitasking ability [6]. The impact of these symptoms on their quality of life, including their psychological well-being, self-esteem, the quality of their relationships and their working life, is so critical that most patients perceive themselves as “chemobrain victims” [7]. Moreover, these symptoms can last longer than 10 years after treatment [8]. Although there is still no validated intervention to manage CRCI, various approaches have been evaluated. Non-pharmacological interventions have shown promising results including cognitive stimulation and physical activity, which seem to be the most beneficial techniques for improving cognitive impairment and quality of life [2,9,10,11,12,13].

Cognitive stimulation is defined as engaging in a series of activities aimed at general cognitive improvement [14]. Among these interventions, the most used are cognitive training and cognitive behavioral therapy. The first is based on neuroplasticity and improvement through repeated and intensive exercise. Exercises are usually adaptive, and their difficulty increases as the patient′s performance improves. The second is based on psychoeducation and enhancement of general cognitive resources. This approach does not target specific cognitive domains but focuses on compensatory strategies and how to transfer these strategies in real life.

Physical activity is defined as any movement of the body (produced by contraction of skeletal muscles) that increases energy consumption [15]. This is an umbrella term that can implies various kinds of training, from cardiovascular fitness to resistance exercise training and pilates.

Recently, it has been proposed that the combination of cognitive stimulation and physical activity could be more beneficial for cognition. The hypothesis is that physical activity works as a “plasticity facilitator”, promoting neurogenesis and cognitive stimulation works as a “guide”, regulating synapse formation [16]. Outside the oncology field, several studies have shown promising results of combined interventions on cognition [16,17,18,19,20,21,22]. Instead, there has been little study of this approach to improving CRCI. To our knowledge, only two pilot studies have tried to investigate the feasibility and efficiency of multimodal interventions [23,24], but their sample is too small to permit conclusions about the efficacy of the intervention. Moreover, both studies proposed a simultaneous combination of physical activity and cognitive stimulation, which could have been too demanding for patients.

According to the American Cancer Society (ASCO) guidelines, published in 2016, primary care clinicians in the event of suspected cognitive impairment should refer patients to a neurocognitive expert for assessment and rehabilitation [25]. Despite these guidelines, to date, clinicians and providers still report being uncertain about the potentially available interventions and how to address CRCI adequately [9,26], and patients′ demand for support is often not addressed [7,12,27].

One explanation for the lack of support to improve CRCI is that these interventions are difficult to set up in routine and supportive care.

The main issue is the unavailability of professionals such as neuropsychologists and physical therapists in hospitals and healthcare centers. When present, these professionals are in small number, and often they are not enough to answer the demand of patients, which may have an effect not only on patients’ quality of life but also the professionals′. Moreover, these interventions are usually not adapted to patients’ needs and schedules, resulting in low adherence and compliance to the intervention [28].

These issues have already been encountered in the healthcare field, and to overcome them, attention has turned to e-health, which can be defined as health services and information delivered through the internet or related technologies [29]. E-health has already proved to optimize general [30] and preventive patient care [31] and to promote physical activity [28] and patient engagement [32,33]. More recently, the use of computerized or digital interventions has proven to be also efficient in the improvement of mental health [32,34,35,36,37].

Computerized interventions can be easily performed at home, using personal mobile devices or laptops, and can be supervised remotely by a professional, making the intervention more affordable and accessible, reaching isolated or stigmatized groups (i.e., patients living in rural areas and patients with special needs) [38]. Additionally, these interventions can be sensitive to the user′s performance and adapt the level of difficulty accordingly to the user′s abilities. They also allow the standardization of the intervention, reducing proficiency bias (inequality in the application of the intervention, due to differences in practitioners or resources of the different sites). Finally, they can reduce time and workload pressures on professionals, minimize their errors, and allow them to support more patients.

However, the efficacy of such interventions is not still clear, and because of the novelty of the subject, there is not yet a systematic review summarizing the state of knowledge on computerized interventions for CRCI. Furthermore, to date, computerized cognitive stimulation and physical activity have not been yet compared. Additionally, previous reviews have focus mostly on breast cancer, and brain cancer has been often overlooked. Thus, this systematic review presents the state of the knowledge on existing computerized cognitive stimulation and physical activity intervention studies to improve CRCI and reports their benefits on objective cognitive functioning as well as subjective cognitive complaints in the entire oncology population.

We believe this work to be essential to help clinicians and researchers during the decision-making process, as it provides a summary of available evidence on computerized cognitive stimulation and physical activity interventions for the improvement of cognition.

## 2. Objectives

This review aimed to analyze the efficacy of the current computerized cognitive stimulation and physical activity interventions on CRCI. To this end, the proposed systematic review aimed at answering the following questions:Are computerized physical activity, computerized cognitive stimulation or combined interventions efficient to improve cancer-related cognitive impairment?What are the risks of bias of the existing studies? Additionally, what is the level of quality of the evidence?

The review questions were defined using the PICO methodology:-Population: Participants living with cancer (adults and children).-Intervention: computerized physical activity, computerized cognitive stimulation and combined intervention.-Comparisons: usual care, wait-list group or any other intervention other than computerized intervention.-Outcome: cognitive functioning.

## 3. Materials and Methods

This systematic review adhered to the PRISMA statement for reporting systematic reviews [39]. All methods were prepared a priori. The protocole of this systematic review has been registred on OpenScience (OSF) (Registration DOI 10.17605/OSF.IO/HW8YD).

### 3.1. Search Strategy

Literature search strategies were developed using Mdical Subject Headings (MeSH) and text words related to cancer, cognitive and physical training. References from PUBMED and Web of Science (WOS) were searched. The inclusion criteria for study selection were as follows: (a) studies in oncology (both adults and children with cancer or cancer survivors); (b) studies on computerized cognitive stimulation or physical activity interventions or combined interventions; (c) assessing their impact on cognition (objective and/or subjective assessment); (d) from January 2000 to December 2020; (e) articles in English.

The authors agreed on the definition of computerized interventions, like all interventions that use technology in some form to provide an interactive, multisensory learning experience. More precisely, all studies delivering computer-based cognitive training were included, whereas all studies proposing video-conference psycho-education were excluded because they were not considered interactive. Concerning computerized physical activity, the authors agreed to exclude studies with intervention including some form of meditation (e.g., yoga) because in these cases it is not clear if the efficacy is related to physical activity or meditation. Computerized physical activity was defined as physical activity delivered through a computerized system.

We excluded letters, comments, surveys, observational studies, and case reports. We decided to include randomized controlled trial protocols for a wider understanding of current and ongoing interventions for cognitive remediation. The following keywords were used in the search algorithm: “cancer,” “tumour,” “malignancy,” “neoplasm,” “lymphoma,” “leukaemia,” “physical activity,” “exercise,” “physical training,” “cognitive stimulation”,” cognitive rehabilitation,” “cognitive intervention,” “cognitive training,” “web-based,” “on-line,” “Internet-based,” “digital,” “gamification,” “e-health” (digital health), “m-health” (mobile health), “Cognit*.” We first developed a strategy for PUBMED search, which once finalized was adopted for the WOS search. The PUBMED search strategy (Appendix A) can be found in the Appendix A.

We also reviewed reference lists and relevant systematic reviews for additional potentially eligible studies.

### 3.2. Selection Process

Two authors (G.B. and M.L.) independently screened the 3589 titles and abstracts of the studies for their relevance in the free-access internet-based software Rayyan QCRI software, using the blind mode [40]. All exclusion criteria have been reported on the software by both authors (G.B., M.L.).

When both authors completed the review of titles and abstracts, the blind mode was disabled and conflicts in the inclusion decision were resolved in agreement by both authors. Eligible articles were exported to Zotero for a second full-article screening, performed independently by both authors.

Any discrepancy in this phase has been discussed by both authors, until full agreement was reached. All reasons for non-inclusion have been recorded.

Twenty studies met all the eligibility criteria for final review. A flow chart detailing the identification of studies is provided in Figure 1.

### 3.3. Data Extraction

Data from selected articles have been summarized in two tables: (1) computerized-cognitive stimulation and (2) physical activity.

For each study, in an excel sheet the first author has manually extracted and reported studies′ data including study design, participants′ information, intervention content and format, expected outcomes and measures, results and conclusions. The second author has later verified the accuracy of the extracted items. Discrepancies and uncertainties about the extraction of data were resolved by the consensus of both authors.

### 3.4. Quality Assessment and Risk of Bias

The risk of bias in included studies was assessed by two authors (GB and SL) using the Rob2 tool [41]. The tool is structured to consider 5 domains that can be a source of bias: randomization process, deviations from intended interventions, outcome data, measurement of the outcome and selection of the reported results. Each domain was assessed as low, high or unclear risk of bias. Upon judgment of the risk of bias for each domain, the authors judged an overall risk of bias for each publication included.

The strength of the body of evidence was rated independently by the two above mentioned authors using the Evidence-based Practice Centers (EPCs) method [42]. This method required an assessment of 4 domains: risk of bias, directness, consistency and precision. For an overall assessment of the body of evidence, EPCs use 4 grades: insufficient, low, moderate and high.

## 4. Results

From an initial 3776 articles identified from the search, duplicates (*n* = 187) and studies not meeting the inclusion criteria based on title and abstract (*n* = 3538) were excluded, resulting in 51 articles as potentially relevant papers and selected for full-text assessment (Figure 1). The main reasons for exclusion were the study design (e.g., no intervention program or non-computerized intervention), the population (e.g., non-cancer population or studies on animal model) and no cognitive outcome (studies non reporting intervention′s effects on cognition). After the full-text assessment, 20 studies were finally included in this review. Among the included studies, sixteen evaluated computerized cognitive stimulations [43,44,45,46,47,48,49,50,51,52,53,54,55,56,57,58] and four evaluated a computerized physical activity program [59,60,61,62]. Thirteen studies were randomized-control studies [44,45,46,47,48,51,54,55,56,57,61,62,63] and seven pilot studies (reporting also efficacy of the intervention) [43,49,50,52,53,58,60]. Most studies have been judged to have an overall high level of risk of bias, three studies arose some concerns regarding the risk of bias [44,48,59] and only one study resulted in having a low risk of bias [46] (Figure 2 and Figure 3). Upon the judgment of risk of bias, the consistency and the precision of studies, the strength of evidence of interventions for cognitive impairment was judged to be low by the authors (Table 1).

### 4.1. Computerized Cognitive Stimulation

Table 2 showed the details of computerized cognitive stimulation studies.

#### 4.1.1. Characteristics of Studies

Design: Among the sixteen computerized cognitive stimulation studies [43,44,45,46,47,48,49,50,51,52,53,54,55,56,57,58], ten were randomized controlled studies [44,45,46,47,48,51,53,55,56,57], and six were feasibility/pilot studies [43,49,50,52,54,58].Population: Five studies included breast-cancer patients [43,46,49,51,56], six did not focus on a specific type of cancer [44,47,50,53,54,55], four included patients with brain cancer [45,48,52,57] and one studied patients with prostate cancer [58]. The intervention was mainly proposed to adults, only two studies proposed the intervention to children [45,50] and none of them concerned older patients (>70 years old). In three studies, chemotherapy treatment was an inclusion criterion [44,47,51]. In one study, intervention was proposed after hematopoietic stem cell transplantation [55], after surgery [57] or to patients on androgen deprivation therapy [58]. The intervention was proposed mostly to cancer survivors [43,44,45,46,51,53,54] and not directly during or after treatment. When stated, the time since treatment completion was between 22 weeks [49] and 6 years [51]. Most of the studies had a control group (14/16), among which 9/16 were wait-list groups.Sample size: The mean sample size was 97 participants (minimum: 16 participants [52]; maximum: 242 [44]).Cognitive evaluation: In most studies [43,44,46,47,48,49,50,51,53,54,55,56,58,64], the efficacy of the intervention was evaluated at the end of the intervention by both objective and subjective cognitive assessment, while in two studies it was evaluated only by objective cognitive assessment [52,57] Twelve studies also investigated the maintenance of the efficacy of the intervention on cognition [43,44,46,48,49,50,52,53,54,55,56,58].

#### 4.1.2. Characteristics of Cognitive Stimulation Programs

Included studies were based on the neuroplasticity model and offered cognitive training to patients. Moreover, sessions of psychoeducation [48,49,53,54], metacognition [57] or individual coaching [55] were sometimes combined with cognitive training. The cognitive training programs used were Brain HQ [49,58], CogMedRM [50], TNP [52], C-Car [48], Happy Neuron Pro [46], Insight [44], Cogmed [45], Aquasnap [43], RehaCom [47] and Lumos Lab Inc [51]. Most of the programs trained multiple domains frequently affected in cancer patients [43,44,46], while others focused on specific domains, like attention/ information processing [48,58], executive functions [51] or memory [45].

Nine programs out of sixteen were home-based with mail or telephone support if needed [43,44,46,50,53,54,58,64,65], with only one study proposing remote psychological supervision [45]. Seven studies proposed an on-site computerized intervention with the direct supervision of a neuropsychologist [47,48,49,52,55,56,57]. The length of the programs ranged from three weeks [55] to 24 weeks [43], with a frequency of one [44,46,52,53,54,55] to five [45,58] sessions per week. Each session lasted from 20 min [51] to two hours [48].

#### 4.1.3. Efficacy of Cognitive Stimulation Programs

Eleven studies out of sixteen showed a post-intervention improvement in objective cognitive domains [45,46,47,48,50,51,52,54,56,57,58], especially memory [45,46,47,48,52,56,57], attention [45,48,54,57], processing speed and executive functions [51,52,56]. Eight studies also reported a significant post-intervention improvement in cognitive complaints [43,44,45,47,48,49,54,56]. Participants or their caregivers (for subjects under the age of 18) also reported improvement in mental fatigue [48,49], cognitive failure [48], planning and task monitoring [51], and a reduction in attention, executive dysfunctions [45], learning problems [50] and emotional distress and sleeping disorders [49]. The beneficial effects of the intervention were maintained at 2-, 3- and 6-month follow-up in five studies [43,44,48,52,56]. When reported (6/16 studies), the participants’ adherence and satisfaction with these interventions were high (compliance between 65 and 95%) [50,52,56,58,64,65]. Table 3 showes the efficacy of cognitive stimulation programs. 

### 4.2. Computerized Physical Activity

Table 4 shows the details of computerized physical activity studies.

#### 4.2.1. Characteristics of Studies

Design: Among the four studies on physical activity programs [59,60,61,62], three were randomized controlled trials [59,60,61] and one was a feasibility study [62].Population: Two studies proposed the intervention to adults between 18 and 70 years old [59,60] while one study focused on the elderly (>70 years old) [62] and another on patients younger than 18 years old [61]. One study was proposed to patients with breast cancer [59], two studies did not focus on any specific type of cancer [60,61] and the intervention was proposed to both breast and prostate cancer patients in the last one [62]. Three studies provided treatment history, which showed that most patients were treated with chemotherapy, hormone therapy and/or radiotherapy [59,60,62]. None of them reported information on the time of completion of treatment. All studies had an active control group. Sample size: mean sample size was 66 participants (minimum: 32 [60]; maximum: 78 [61,62]).Cognitive evaluation: all studies assessed the efficacy of the physical activity intervention on objective cognitive functioning, and two studies also on cognitive complaints [60,62]. Only one study evaluated the cognitive effect after the post-intervention assessment with another assessment at 6 months [59].

#### 4.2.2. Characteristics of Programs 

All studies proposed a physical activity program composed of aerobic exercises, except one study which does not specify the type of physical activity proposed [61]. In the latter, participants in the intervention group received educational materials, an active monitor and access to a website where they uploaded their physical activity data. Based on their daily activity levels, patients accumulated points which allowed them to progress through levels of the website and to gain small prizes. Three studies out of four proposed a home-based intervention [59,60,61], and only one of them included remote supervision [60]. The remaining articles gave access to the software but did not supervise patients′ compliance to the program.

The length of the program ranged from 4 weeks [62] to 24 weeks [61], with a frequency ranging from open access [61], to once a week [62] and three times a week [59,60]. When reported, the length of each session ranged from 5 min [62] to 90 min [59].

#### 4.2.3. Efficacy of Programs

The authors of two studies reported an improvement in attention, information processing speed, verbal memory and executive functions [59,60]. One of those studies also reported a significant improvement in cognitive complaints [60]. In the only study with follow-up assessments [59], the beneficial effects of the intervention on cognition were maintained at the 6-month follow-up. Only half of the studies reported the adherence rate of participants in the intervention program [59,61], with rates of adherence higher than 80%. Table 3 summarizes the efficacy of studies on objective and subjective cognition. Table 3 showes the efficacy of physical activity programs.

## 5. Discussion

Interest in interventions to support patients with CRCI has recently increased, resulting in a growing body of literature showing promising results, which sometimes can be inconsistent and confusing for healthcare providers. This review aimed to present and summarize the state of knowledge of the efficacy of computerized cognitive stimulation and physical activity in reducing CRCI. Other reviews and meta-analyses have already investigated the efficacy of pharmacological and non-pharmacological interventions on cognition, but, to our knowledge, none of them focused on computerized intervention in the oncological population [2,10,66,67]. These previous reviews reported cognitive interventions and physical activity to have beneficial effects on cognition, but the applicability of these interventions in hospitals and healthcare centers has been questioned. They propose rigid schedules, and adherence of patients to these interventions is often modest. For these reasons, it was decided to focus the review of the literature only on computerized intervention, which seems to address these limits.

However, findings from this review are consistent with the previous literature, indicating the beneficial effects of computerized cognitive stimulation (11/16 studies) and physical activity intervention (2/4 studies) on cognition. Improvement was mainly found in cognitive complaints, memory, attention, processing speed and executive functions.

Results in this review need to be interpreted taking into consideration the limits of the studies and the overall strength of evidence.

The first factor to consider is the novelty of the subject with the consequent lack of studies, especially regarding physical activity and multimodal interventions. Between 2000 and 2020, there were only sixteen studies on computerized cognitive stimulation and four on computerized physical activity.

Secondly, the authors of this review have limited confidence in the findings of the included studies, especially because of their statistical, methodological and clinical heterogeneity. For example, in most of the included studies, the intention-to-treat analysis was not performed, which could have greatly impacted the reported results. Additionally, in several studies assessors for the cognitive test were not blinded, which could have resulted in biased results. Studies varied also concerning the characteristic of the control group (in some studies the intervention group was compared only to the waiting list group).

Another reason for concern is the variability across studies of cognitive assessment (tests and questionnaires), the domain of cognition evaluation and timing of assessment. For example, Bellens et al. [43] assessed only perceived cognition and did not perform objective cognitive assessments, whereas Galliano-Castillo et al. [59] used only two tests to evaluate effects on short-memory, attention and processing speed information, without investigating the perceived cognition. Finally, in physical activity studies, cognitive improvement was not always the primary outcome and it was not always fully investigated, and most importantly, intervention may have not been shaped with this aim. Concerning cognitive stimulation, studies reported also an important variability concerning the cognitive domains trained; indeed, some programs like the HappyNeuron software can train various aspects of cognition, while others like the Cogmed software train only working memory and cognition. This difference between interventions makes it difficult to state the general efficacy of computerized cognitive interventions. Additionally, none of this intervention has been specifically developed to train impacted domains of cognition or to be adapted to these patients’ characteristics. For example, while it is well known that one of the main concerns of younger patients with CRCI is the return to work, none of these interventions have targeted exercises to train domains of cognition that can more impact work performances. Future research should also develop interventions to help patients return to work, proposing various scenarios to adapt to patients working experiences.

This review revealed also the lack of research on the efficacy of a multi-approach combining both types of intervention in patients with cancer. Only two small pilot studies [23,24] were identified in the search, but they were not included because only cognitive stimulation was computerized. Moreover, these studies had several limits, such as small samples (10 and 28 patients) and concerns concerning the design of the intervention. The body of evidence was judged insufficient and consequently, authors of this review could not draw any conclusion on the efficacy of such type of intervention. Outside research in oncology, the limits of mono-therapeutic approaches have already been enlightened and it has been suggested that a combination of physical activity and cognitive stimulation could have better results than the two approaches alone. Moreover, in several studies, this hypothesis has been proven to be correct [16,18,19,20,21,22]. Because the cognitive impairment in oncological patients is multi-causal, it seems possible that a multi-approach intervention, acting on different aspects of the cognition, may yield better results. Thus, further investigation is needed to investigate the efficacy of such an intervention for CRCI.

Computerized interventions for CRCI have resulted to be mainly focused on adults, while there are few interventions proposed to children (only 3 studies) and elders (only one study). The causes of the lack of computerized interventions can be easily imagined when it comes to the elderly population. Indeed, elders are frequently reported to lack familiarity and interest toward technologies [67,68]. Despite, these preconceptions, older adults have shown to benefit from these kinds of interventions, even more than from paper-and-pencil cognitive training [69]. Upon these results, further studies are needed to investigate the acceptability and feasibility of such interventions among the elderly population.

It is more difficult to imagine the reasons for the lack of computerized interventions for the oncological pediatric population. Being used to video games and technologies, it seems logical to propose interventions for children using these tools, which seem closer to their interests and more adapted to their needs and abilities. Furthermore, outside the oncology field, computerized interventions have shown to be very efficient for several areas in the pediatric population [70,71,72,73,74,75,76].

In this case, it does not seem very useful to conduct feasibility studies, but further investigation to verify the efficacy of computerized interventions in the pediatric oncology population is needed.

Two different approaches for computerized cognitive stimulation were identified: cognitive training [43,44,45,46,47,51] and cognitive behavioral therapy [48,53,54]. As explained in the introduction, the two interventions focus on two aspects of cognition: the brain’s ability to change and adapt thanks to experiences and training, and the ability to learn new strategies or new abilities to compensate for a deficit or weakness in a specific area. While both interventions are helpful for cognitive improvement, cognitive training is easier to propose, because it does not require the presence of a psychologist and allow brief intervention (around 20 min). Nevertheless, the combination of both approaches could be the best method to improve cognitive functioning, as demonstrated by non-computerized cognitive interventions [12,13]. We identified only one study combining both approaches [48]: it reported relevant improvements in cognitive complaints, attention and verbal memory compared to a wait-list group. Further studies with active control groups are needed to confirm its superiority over single cognitive training or cognitive behavioral therapy.

The computerized physical activity interventions identified in this review were all based on one approach, which was resistance and aerobic exercises, with a heterogeneous intensity. Among the non-computerized physical activity interventions explored, aerobic exercises alone or in combination with resistance training already proved to be more effective for the cognitive improvement of cancer patients [77,78], while low-intensity exercise, like walking exercises [79] and physical activity coupled with meditation [80,81,82,83,84], showed contrasting results. A recently published review [85] comparing different types of physical interventions did not report any differences between aerobic exercises, combined aerobic and resistance exercises, or mind–body interventions. As the authors of this review recommend, there is still a need for further studies to identify the optimal physical intervention to improve CRCI.

When reported, compliance in both computerized interventions was consistently high, ranging from 65 to 95%. This promising result can be explained by the adaptability of computerized interventions to patients’ needs and schedules, along with their capacity to improve engagement and motivation through rewards. Computerized interventions can also address the issue of the absence of trained professionals in hospitals and healthcare structures, allowing healthcare centers to share their trained professionals remotely. Furthermore, during the COVID-19 pandemic, it has become increasingly essential to develop web-based interventions to ensure the continuity of care.

However, to maximize adherence and the effectiveness of the intervention, facilitators should be included, as has been already suggested by Kim et al. [85] in their systematic review. According to this review, except for the study of Kesler et al. [51] which reported high rates of compliance even without supervision, having a health care worker or psychologist supervising the intervention and supporting patients during the program proved to be beneficial.

Future studies should also assess the cost effect of computerized interventions, as has already been suggested by Von Ah et al. [86]. This would permit in the future to better understand the applicability of such interventions in healthcare routines.

Limitations of this study included heterogeneity of the included studies (study design, sample, cognitive assessment, etc.); because of that, a meta-analysis could not be performed. Results could have been affected by selection bias because selection and ranking were performed independently by two authors. Furthermore, the limited number of physical activity interventions and combined interventions may have affected the interpretation of results. The authors decided to not search for grey literature and, as such, non-published unfinished studies or studies reporting negative results could have been overseen.

## 6. Implication for Practice

As cited above, health care professionals are struggling to propose rehabilitation alternatives requested by cancer patients. One reason is the absence of an updated summary of the most recent and modern interventions and their efficacy in the improvement of cognitive dysfunction. Furthermore, the majority of the existing reviews are focused only on breast cancer survivors, leaving a large proportion of cancer patients uncovered. To our knowledge, our work is the first systematic review to synthesize and compare the results between computerized-cognitive intervention and physical activity which includes patients with different tumours of all ages. Therefore, this review, which is not focused on a specific subgroup of cancer patients, taking into consideration patients of different ages and sex, has the advantage to give a wider updated overview of computerized cognitive rehabilitation. Finally, this review contributes to updating the state of knowledge on digital technologies, which is extremely important, given the rapid progress of new technologies.

## 7. Conclusions

This review proposes a summary of the latest research in interventions in CRCI, to permit care professionals to have a clearer view of what intervention can be proposed to patients, and to researchers to take inspiration for their next studies’ design. In this review, it has been demonstrated the benefit of computerized cognitive stimulation and computerized physical activity on CRCI. However, further studies are needed that directly compare each of these interventions on cognition. Furthermore, the added value of a combined intervention including computerized cognitive stimulation and physical activity should be investigated so that the best supportive care may be offered to cancer patients with CRCI in clinical practice.

## Figures and Tables

**Figure 1 cancers-13-05161-f001:**
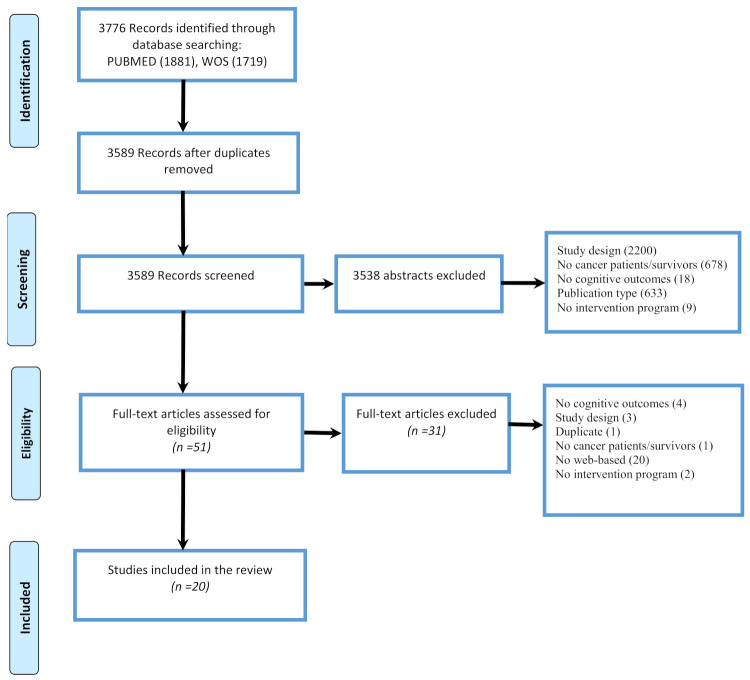
PRISMA flowchart of the identification and inclusion of studies.

**Figure 2 cancers-13-05161-f002:**
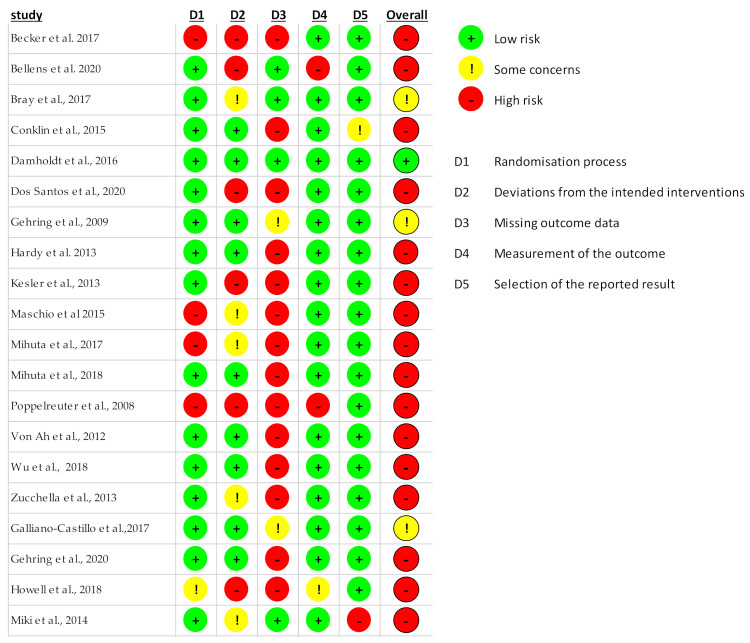
Detailed risk of bias judgment for each domain of the Rob2 tool.

**Figure 3 cancers-13-05161-f003:**
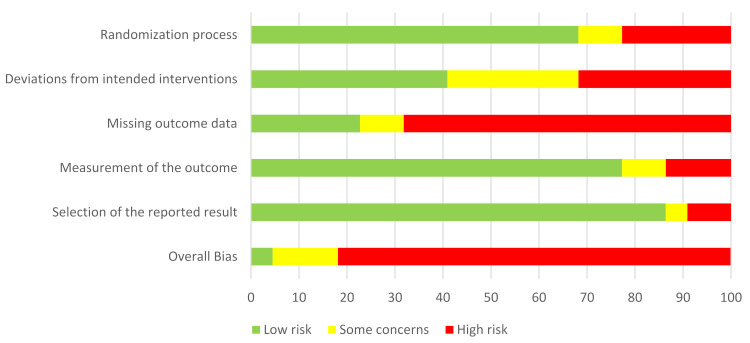
The overall risk of bias judgment using the Rob2 tool.

**Table 1 cancers-13-05161-t001:** Summary of strength of evidence for cognitive outcome.

Intervention	N of Randomized Studies/N of Pilot Studies	Strength of Evidence	Risk of Bias	Directness	Consistency	Precision
Computerized cognitive stimulation	10/6	Low	High	Direct	Inconsistent	Imprecise
Computerized physical activity	3/1	Low	High	Direct	Unknown	Imprecise

**Table 2 cancers-13-05161-t002:** Computerized-cognitive stimulation studies.

Publication	Study Design and Assessment	Participants	Intervention	Supervision	Outcomes and Tools	Results	Conclusion
Becker et al., 2017 [49]	Pilot study Pre-intervention Post-intervention Two-weeks after intervention	Breast cancer survivors N = 20	3–4 times/week of 45-min cognitive training (home-based) + 6 90-min group classes supervised by a nurse (BrainHQ) 6 weeks of the program	Home-based cognitive training without supervision On-site group intervention supervised by a nurse	Efficacy: cognition (CVLT; COWAT; SDMT; PROMIS; MMQ) and quality of life (FACT-G)	Efficacy: no significant results	Program feasible but no significant cognitive improvement
Bellens et al., 2020 [43]	Pilot RCTBefore intervention Cognitive assessment every 4 weeks For other measures: after 3 and 6 months	Breast cancer survivors early intervention group: *n* = 23 Delayed intervention group: *n* = 23	3×/week of 60-min video-game over 6 months (Aquasnap)	Home-based intervention with 3 monthly meetings with the principal investigator. Contacts by email or telephone of study team if problems	**Primary outcome**: MyCQ (cognitive tests) **Secondary outcomes**: Anxiety (HADS), self-reflectiveness (BCIS), cognitive complaints (CFQ), quality of sleep (PSQI)	**Primary outcome**: non-significant change in overall MyCQ score between groups for baseline to 3 months. **Secondary outcomes**: Significant improvement in CFQ scores (*p* = 0.029)	Improvement of cognitive complaints
Bray et al., 2017 [44]	RCT T1: before intervention T2: after intervention T3: 6 months later	Cancer survivors Intervention group: *n* = 121 Control group: *n* = 121	40-min sessions/week 15 weeks intervention (Insight) or standard care	Home-based intervention without supervision or support	**Primary outcome**: FACT-COG PCI (cognitive complaints) **Secondary outcome**: cognitive tests (Cog-State), anxiety/depression (General Health Questionnaire); QoL (FACT-G); fatigue (FACT-F) and stress (Perceived Stress scale)	**Primary outcome**: Difference between the groups on the FACT-COG PCI was statistically significant, with less PCI in the intervention group at T2 (*p* < 0.001) sustained at T3 (*p* < 0.001) **Secondary outcomes**: No significant difference between groups in cognitive tests at T2 and T3 but significant improvement at T2 on anxiety/depression, fatigue and stress.	Improvement of cognitive complaints
Conklin et al., 2015 [45]	RCT pre-intervention post-intervention	Survivors of childhood acute lymphoblastic leukaemia (ALL) or brain tumour (BT) Intervention group: *n* = 34 Waitlist group: *n* = 34	25 training 30–45 min sessions (Cogmed)+ weekly telephone-based coaching over 5 to 9 weeks	Home-based intervention with weekly coaching telephone calls	**Primary outcome**: spatial span backwards (WISC-IV) **Secondary outcome**: Other cognitive tests (WISC-IV, CPT-II, WJ-III) and parent-reported measures (CPRS-3, BRIEF)	**Primary outcome**: Greater improvement of spatial span backward in the intervention group than the control group (*p* = 0.002) **Secondary outcomes**: Greater improvement of intervention vs. control group in WM, attention and processing speed (*p* = 0.01). Improvement in reported-attention and executive dysfunctions (*p* < 0.01)	Intervention feasible and efficacious for childhood cancer survivors
Damholdt et al., 2016 [46]	RCT pre-intervention post-intervention 5-month follow-up	Breast cancer survivors Intervention group: *n* = 94 Waitlist group: *n* = 63	30 training 30 min-session over 6 weeks (HappyNeuron Pro) with phone support	Home-based intervention with telephone and email support in case of difficulties. 2 phone calls: one at the beginning of the program and one at the end.	**Primary outcome**: PASAT (attention and working memory) **Secondary outcomes**: Cognitive complaints (CFQ), verbal learning (RAVLT), WM (WAIS-IV), EF (Multilingual aphasia examination; D-KEFS, Cognitive estimation task)	**Primary outcome**: no significant time x group interaction for PASAT **Secondary outcomes**: post-intervention and 5-month follow-up significant increase in verbal learning (*p* = 0.043) and digit span backwards (*p* = 0.040) for the intervention group compared to a waiting list	Improvements in verbal learning and working memory including at 5-month follow-up
Dos Santos et al., 2020 [47]	RCT pre-intervention post-intervention	Cancer patients The experimental group (A): N = 55 Control group B: N = 56 Control group C: N = 56	Experimental group: computer-assisted cognitive rehabilitation (Rehacom). 9 sessions (45–60 min) over 3 months Group control B: cognitive exercises at home (booklet) 9 sessions (30–60 min) over 3 months Group control C: a phone call. 9 over 3 months	On-site intervention supervised by a neuropsychologist	**Primary outcome**: PCI FACT-COG **Secondary outcomes**: episodic memory (Grober and Buschke), attention (d2), executive functions and processing speed (Verbal fluency test, TMT), working memory and short-term memory (digit span WAIS-IV)	**Primary outcome**: no significant difference between groups on PCI improvement **Secondary outcomes**: compared to control groups significant improvement of PCI (*p* = 0.02), perceived cognitive abilities (*p* > 0.01) and working memory (*p* = 0.03) for group A	Improvement of cognitive complaints and working memory
Gehring et al., 2009 [48]	RCT Pre-intervention post-intervention 6-month follow-up	Adult patients with gliomasIntervention group: *n* = 66 Waitlist group: *n* = 69	6 weekly 2h-sessions of attention program (C-Car) and psychoeducation	On-site intervention with the supervision of a neuropsychologist	Attention (SCWT, DS, LDST, MST, TEA), verbal memory (WLT), executive function (CST, LF, BADS) and cognitive complaints (MOS CFS, CFQ), fatigue (MFI)	Cognitive tests: significant group differences for attention (*p* = 0.028) and verbal memory (*p* = 0.015). Cognitive complaints: significant group difference overtime for CFS total score, burden, and CFQ total score (*p* = 0.003) and mental aspects of fatigue (*p* = 0.049)	Improvement of cognitive complaints, attention and verbal memory
Hardy et al., 2013 [50]	Pilot study Pre-intervention Post-intervention 3 month follow-up	Survivors of childhood cancer Intervention group *n* = 13 Control group *n* = 7	5–8 weeks 25 sessions (Cogmed RM)	Home-based intervention with phone-based coaching support	Preliminary efficacy of the program (WASI, WRAML2, Conner’s rating scale, SERS)	Efficacy: significant group differences for working memory (*p* = 0.05) and parent-reported learning problems (*p* = 0.05)	Program feasible and Improvement of working memory and learning abilities
Kesler et al., 2013 [51]	Feasibility study pre-intervention post-intervention	Breast-cancer survivors Intervention group: *n* = 21 Wait list: *n* = 20	48 session (20-30 min) of EF program (Lumosity) over 12 weeks	Home-based intervention without supervision and reminder	**Primary outcome**: WCST (flexibility) **Secondary outcome**: EF (letter fluency test, BRIEF), verbal memory (HVLT-R), working memory (digit span), processing speed (symbol search) and depression (CAD)	**Primary outcome**: significant improvement of flexibility in the intervention vs. control group (*p* = 0.008) **Secondary outcomes**: Improvement in letter fluency and symbol search (*p* < 0.01)	Improvement of flexibility, letter fluency and symbol search
Maschio et al., 2015 [52]	Pilot study Pre-intervention After intervention 6-month follow-up	Brain-tumor patients Intervention group *n* = 16	1h/week 10 weeks RehabTr	On-site intervention With the supervision of a neuropsychologist	Cognitive improvement (MMS; TMT; frontal assessment battery; Raven Matrices; ROCF-Copy and recall; Clock Drawing test; Span forward and backward; 15 Rey-Osterrieth Word list, fluency test	Primary outcome: significant improvement of memory (*p* = 0.0017; *p* = 0.036) and fluency (*p* = 0.043)	significant improvement of memory and fluency after intervention and at 6-month follow up
Mihuta et al., 2017 [54]	Pilot study pre-intervention post-intervention 3-month follow-up	Cancer survivor Cancer intervention group: *n* = 13 Non-cancer intervention group: *n* = 21 Non cancer wait-list: *n* =17	4-week 2-h session (eRECog program)	Home-based intervention with reminder emails	**Primary outcome**: PCI FACT-Cog (cognitive complaints) **Secondary outcome**: other cognitive complaints questionnaires (BAPM, BADL, EORTC QLQ-C30, IADL), cognitive tests (WebNeuro), distress (K10), illness perception (BIPQ) and program satisfaction	**Primary outcome**: No significant interaction for PCI **Secondary outcome**: Significant improvement of cognitive complaints (BADL) and attention in the intervention group	High participant satisfaction and some improvements in subjective and objective cognitive functioning
Mihuta et al., 2018 [53]	RCT pre-intervention post intervention 3-month follow-up	Cancer patients Intervention group: *n* = 40 Waitlist group:*n* = 36	30–60 min sessions/week 4 weeks (eRECog program)	Home-based intervention with emails reminder 5 days after non-completion of the session and phone call after 3 mails reminder without answer. Encouragement mail after completion of the first module	**Primary outcome**: PCI FACT-Cog (cognitive complaints) **Secondary outcome**: other cognitive complaints questionnaires (BAPM, BADL, EORTC QLQ-C30, IADL), cognitive tests (WebNeuro), distress (K10), illness perception (BIPQ), fatigue (EORTC QLQ-C30) and program satisfaction	**Primary outcome**: No significant interaction for PCI **Secondary outcome**: Significant improvement of the prospective memory IADL score in the intervention vs. control group. No significant interaction for other variables	No significant group effect on cognition
Poppelreuter et al., 2008 [55]	RCT Pre-intervention Post intervention 6-month follow-up	Patients after HSCT Intervention group (NPT) *n* = 21 Intervention group (PC) *n* = 26 Control group *n* = 28	1 h/week 3–5 weeks Different training software	On-site intervention NPT = neuropsychological training group (max 8 participants) supervised by an occupational therapist PC = individualized computer-based training + individual coaching	Attention, memory (battery of standardized tests) and cognitive complaints questionnaires (EORTC; MFI; FEDA)	No significant results	No significant improvement
Von Ah et al., 2012 [56]	RCT Pre-intervention Post-intervention 2-month follow-up	Breast cancer survivors Memory training group *n* = 29 Speed of processing intervention group *n* = 30 Wait-list control group *n* = 29	10 1 h sessions over 6–8 weeks (Insight program)	On-site group intervention supervised	Primary outcomes: Objective memory (AVLT; Rivermead Behaviourall Paragraph Recall Test) and speed of processing (UFOV) Secondary outcome: Perceived cognitive functioning (FACT-COG); symptoms distress (CES-D; STAI-S; FACT-F) and quality of life (QOL-CS; SF-36)	Primary outcomes: significant improvement of immediate and delayed memory (*p* = 0.036, *p* = 0.013) at the 2-month follow-up in the intervention group vs. control group. Significant improvement of processing speed after the intervention (*p* = 0.007) and at the 2-month follow-up (*p* = 0.004) Secondary outcomes: significant improvement of perceived cognition (p ≤ 0.005)	Improvement of objective and perceived cognition
Wu et al., 2018 [58]	Pilot study Pre-intervention Post-intervention 8-weeks follow-up	Prostate cancer patients Intervention group (CCT) *n* = 40 Wait-list group: 20	1 h/day, 5 days/week for 8 weeks (BrainHQ)	Home-based intervention with e-mail reminders and weekly phone calls	Efficacy: Objective cognitive functioning (CNS Vital Signs); Self-reported cognitive functioning (PAOFI); Neurobehavioral functioning (FrSBe)	Significant improvement of reaction time in the intervention group vs. control group	Program feasible with some effects on reaction time
Zucchella et al., 2013 [57]	RCT Pre-intervention Post-intervention	Patients with brain tumor Rehabilitation group *n* = 30 Wait-list group *n* = 32	4 weeks 4 1 h sessions/week (training di riabilitatione cognitive; una palestra per la mente)	On-site intervention with direct training and metacognitive training supervised by two neuropsychologist	Cognitive functioning (MMS, digit span, Corsi′s test, RAVLT, PM47, FAB, TMT, ENPA)	**Significant improvement in all the neuropsychological measures in the intervention vs. control group**.	Significant cognitive improvement

AWMA: Alloway Working Memory Assessment; AVLT: Rey Auditory Verbal Learning Test; BAPM: Brief Assessment of Prospective Memory; BCIS: the Beck Cognitive Insight Scale; BIPQ: the Brief Illness Perception Questionnaire; BRIEF: Behavior Rating Inventory of Executive Function; CAD: Clinical Assessment of Depression; CEQ: The credibility/expectancy questionnaire; CF: Category Fluency; CES-D: 20-item Center for Epidemiologic Studies Depression Scale; CFQ: The Cognitive Failures Questionnaire; COWAT: Controlled Oral Word Association Test; CPRS-3: Conner’s Parent Rating Scale ; CPT-II : Conners′ Continuous Performance Test; CST: Concept Shifting Test; CVLT: California Verbal Learning Test; DART: Dutch Adult Reading Test; DASS-21: Depression Anxiety Stress Scales-21; D-KEF: Delis- Kaplan executive function system; DMT: Drie-Minuten Test; EF: Executive Functions; E.N.P.A.: Esame neuropsiclogico per l′afasia; EORTC QLQ-C30: the cognitive functioning scale from the European Organisation for Research and Treatment of Cancer—Quality of Life Questionnaire ; FAB: Frontal assessment battery; FACIT-F: Functional Assessment of Chronic Illness Therapy-Fatigue; FACT-COG: Functional Assessment of Cancer Therapy-Cognitive; FACT-G: self-reported function FACT-General; FEDA: distractibility and retardation in mental task f-MRS: Functional magnetic resonance spectroscopy; FrSBe: Frontal Systems Behavior Scale; HADS: Hospital Anxiety and Depression Scale; HSCT: hematopoietic stem cell transplantation; HVLT-R: Hopkins Verbal Learning Test Revised; IADL: instrumental activities of daily living; K10: Kessler Psychological Distress Scale; LDST: Letter Digit Substitution Test; MIA-A: Metamemory in Adulthood-Anxiety scale; MFI: mental fatigue; MMQ: Multifactorial Memory Questionnaire; MMS: Mini Mental State; MST: Memory Scanning Test; PAOFI: Patient Assessment of of Own Functioning Inventory; PASAT: Paced Auditory Serial Addition Test; PCA: Perceived Cognitive Abilities; PCI: perceived cognitive impairment; PedsQLTM: Pediatric Quality of Life Inventory; PM47: Raven′s coloured Progressive Matrices; PROMIS: Patient-Reported Outcomes Measurement Information System; QOL: quality of life; QOL-CS: 41-item Quality of Life-Cancer Surviovrs; RAVLT: Rey Auditory Verbal Learning Test; RCT: randomized control trial; ROCF: Rey-Osterrieth Complex Figure; SCW: Stroop Color-World Test; SDMT: Symbol Digit Modalities Test; SERS: Side Effects Rating Scale; SSMQ: 18-item Squire Subjective Memory Questionnaire; STAI-S: 20-item Spielberg State Trait Anxiety Inventory-State Subscale; TBANS: telephone-based assessment of neuropsychological status; UFOV: Useful Field of View; VVLT: Visual Verbal Learning Test; WASI-II : Wechsler Abbreviated Scale of Intelligence, 2nd Edition; WCST : Wisconsin card sorting test; WISC-IV: Wechsler Intelligence Scale for Children; WJ-III: Woodcock–Johnson Tests of Cognitive Abilities; WM: Working Memory; WRAML2: wide range assessment of memory and learning.

**Table 3 cancers-13-05161-t003:** Summary of the efficacy of computerized-cognitive and physical activity interventions on cognition.

Interventions	Memory	Attention	Executive Functions	Processing Speed	Subjective Cognition
*Computerized-Cognitive Stimulation*					
Becker et al., 2017 [49]					
Bellens et al., 2020 [43]					✓
Bray et al., 2017 [44]					✓
Conklin et al., 2015 [45]	✓	✓		✓	✓
Damholdt et al., 2016 [46]	✓				
Dos Santos et al., 2020 [47]	✓				✓
Gehring et al., 2009 [48]	✓	✓			✓
Hardy et al., 2013 [50]	✓		NA	NA	NA
Kesler et al., 2013 [51]			✓	✓	
Maschio et al., 2015 [52]	✓				*NA*
Mihuta et al., 2017 [54]		✓			
Mihuta et al., 2018 [53]					
Poppelreuter et al., 2008 [55]					
Von Ah et al., 2012 [56]	✓	NA	NA	✓	✓
Wu et al., 2018 [58]					
Zucchella et al., 2013 [57]	✓	✓			NA
*Computerized-Physical Activity*					
Galliano-Castillo et al.,2017 [63]		NA	NA		NA
Gehring et al., 2020 [60]	✓	✓	✓	✓	✓
Howell et al., 2018 [61]	NA	NA		NA	NA
Miki et al., 2014 [62]	NA	NA	✓	NA	NA
*NA = not assessed*					

**Table 4 cancers-13-05161-t004:** Computerized physical activity studies.

Publication	Study Design/Assessment	Participants	Intervention	Supervision	Outcomes/Tools	Results	Conclusion
Galliano-Castillo et al., 2017 [59]	RCT Pre-intervention Post-intervention 6-month follow-up	Breast cancer survivors Intervention group: *n* = 39 Control group: *n* = 37	3 sessions/week (90 min) over 8-week internet-based tailored exercise program	Home-based intervention with individual supervision through instant messages, video-conference sessions and phone calls	6-min walk test (functional capacity) and ACT + TMT (cognitive tests)	**Functional capacity**: significant improvement in the intervention vs. control group for the 2 follow-up assessments (*p* = 0.001) **Cognitive function**: significant improvement on 1/5 of the ACT score in the intervention vs. control group for the 2 follow-up assessments (*p* < 0.05). No effect on TMT scores	Some improvement in functional performance and cognition
Gehring et al., 2020 [60]	RCT Baseline Post-intervention	Stable patients with grades II/III glioma Exercise group: *n* = 21 Control group: *n* = 11	6-month intervention with 3 aerobic sessions/week (20–45 min)	Home-based remotely coached intervention	Attention (SCWT-int; LDST; WAIS-R digit span; test of everyday attention), memory (VVLT; WMS-III verbal paired associates); executive function (CST-Shift; GIT letter fluency, GIT category fluency); cognitive complaints (CFS cognitive functioning scale; CFQ); fatigue, sleep, mood and QoL (MFI; PSQI; POMS; QLQ-BN20; SF-36)	Better post-intervention scores of the exercise group: attention, processing speed, verbal memory, executive function and cognitive complaints	Improvement in several domains of cognition and cognitive complaints
Howell et al., 2018 [61]	RCT Pre-intervention Post-intervention	Adolescent cancer survivors Intervention group, *n* = 53 Control group, *n* = 25	24 weeks web-delivered physical activity intervention	Home-based intervention without supervision	Physical activity (wGT3X-BT, ACTi Graph); fitness; general intelligence (vocabulary and visual-spatial construction WASI) and flexibility (Delis-Kaplan Executive Function System); quality of life (PedsQL)	No statistical difference between groups for mean change in weekly MVPA. No significant improvement in intervention vs. control group for cognitive scores	No significant difference between groups on cognition
Miki et al., 2014 [62]	Feasibility study pre-intervention post-intervention	Breast and prostate elderly cancer patients Intervention group: *n* = 38 Control group: *n* = 40	4-week 1/week on-site intervention (5min) with a bicycle ergometer	On-site intervention supervised by a therapist	FAB (executive function), BI+IADL (activities of daily living), FACT-G (QoL)	Significant effect of group, time and time x group for FAB score	Feasible intervention to improve cognition

ACT: Auditory Consonant Trigrams; AWMA: Alloway Working Memory Assessment; BAI: Beck Anxiety Inventory; BAPM: Brief Assessment of Prospective Memory; BCPT: Breast Cancer Prevention Trial; BI: The Barthel Index; BIPQ: the Brief Illness Perception Questionnaire; BRIEF: Behavior Rating Inventory of Executive Function; BVMT-R: Brief Visuospatial Memory Test—Revised; CAD: Clinical Assessment of Depression; CEQ: The credibility/expectancy questionnaire; CEZS-D: the Center for Epidemiological Studies Depression; CF: Category Fluency; CFQ: The Cognitive Failures Questionnaire; CF- MOS cognitive functioning scale; COWAT; Controlled Oral Word Association Test; CPET: symptom-limited maximal cardiopulmonary exercise test; CPRS-3: Conner′s Parent Rating Scale; CPT-II: Conners’ Continuous Performance Test; CST: Concept Shifting Test; CVLT-II: the California Verbal Learning Test, 2nd Edition; DART: Dutch Adult Reading Test; DASS-21: Depression Anxiety Stress Scales-21; D-KEFS: Delis- Kaplan executive function system; DMT: Drie-Minuten Test; EF: Executive Functions; EORTC QLQ-C30: the cognitive functioning scale from the European Organisation for Research and Treatment of Cancer—Quality of Life Questionnaire; FAB: Frontal Assessment Battery; FACIT-F: Functional Assessment of Chronic Illness Therapy-Fatigue; FACT-COG: Functional Assessment of Cancer Therapy-Cognitive; FACT-G: self-reported function FACT-General; FACT-P: the Functional Assessment of Cancer Therapy Prostate Module; f-MRS: Functional magnetic resonance spectroscopy; GIT: letter fluency and category; HVLT-R: Hopkins Verbal Learning Test Revised; IADL: Lawton and Brody Instrumental Activities of Daily Living; K10: Kessler Psychological Distress Scale; KKG: German 21item questionnaire Assessment of health and sickness locus of control; LDS: Letter Digit Subsitution Test; MDASI: MD Anderson Symptom Inventory; MFI: Multidimensional Fatigue Inventory; MIA-A: Metamemory in Adulthood-Anxiety scale; MST: Memory Scanning Test; NART: National Adult Reading Test–Revised; NCCN: National Comprehensive Cancer Network; NIH: The National Institutes of Health; PASAT: Paced Auditory Serial Addition Test; PCI: perceived cognitive impairments; PedsQLTM: Pediatric Quality of Life Inventory; POMS: Profile of Mood states; PRMQ: Prospective Retrospective Memory Questionnaire; PROMIS: Patient-Reported Outcomes Measurement Information System; PSQI: Pittsburgh Sleep Quality Index; QLQ: Brain-cancer specific HRQL questionnaire; QOL: quality of life; RAVLT: Rey auditory verbal learning test; RCT: randomized control trial; SCWT: Stroop Color-World Test; SES: the Rosenberg Self-Esteem Scale; SF-36: Medical Outcomes Study Short Form Health Survey Energy Scale; STAI: the Spielberger State-Trait Anxiety Inventory; TBANS: telephone-based assessment of neuropsychological status; TMT: trail making test; VVLT: Visual Verbal Learning Test; WASI: Wechsler Abbreviated Scale of Intelligence; WCST: Wisconsin card sorting test; WISC-IV: Wechsler Intelligence Scale for Children; WJ-III: Woodcock–Johnson Tests of Cognitive Abilities; WM: Working Memory; WMT: Word Memory Test.

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
