# Peer review of "Management of Cancer-Related Cognitive Impairment: A Systematic Review of Computerized Cognitive Stimulation and Computerized Physical Activity"

_cancers, 2021, doi:10.3390/cancers13205161_

Round 1

Reviewer 1 Report

Please provide operational definitions for “Cognitive stimulation” and “cognitive training”. There is some description of this in the discussion, but would suggest moving this to the introduction in order to better orient the reader.

The PICO elements are not clearly reflective of the search. Would suggest that Intervention = computerized physical activity, computerized cognitive stimulation and combined intervention. “Comparison” should reflect the comparison arms that were of interest, for example usual care, waitlist control, or any other intervention. It seems in this study that any comparison group was accepted. “Outcome” should be “cognitive functioning” or “cognition”.

Materials and Methods – It is stated that authors decided not to register the protocol. Suggest to either describe why this decision was made, or just state that “the protocol was not registered”.

Search Strategy item c)  - recommend “assessing their impact on cognition”, rather than “benefit”

Throughout the methods, suggest using past tense to describe what was done, not what will be done (e.g., “we also reviewed references…” rather than “we will review references…”)

For clarity to the reader, I suggest moving the description of included cognitive training and physical activity interventions, to the section describing study eligibility criteria.

Figure 3 – The title contains a typo (“percentage”) but also does not seem to be illustrative for this figure.  Would also suggest re-organizing the figure to align with the order presented in Figure 2 (e.g., Randomisation process in the top row, followed by Deviations from intended interventions, etc.).

Characteristics of computerized cognitive stimulation studies, re: Design. It is stated that there are 10 RCTs and six feasibility/pilot studies.  This conflicts with column 1 in Table 1, indicating 12/4. Some clarification to orient the reader is required.

Characteristics of computerized cognitive stimulation programs. While the program names are listed, providing information on the nature of these programs (i.e., what training elements are included) will better help the reader understand what evidence is available. Linking it back to an operational definition of cognitive stimulation (see comment 1) would be most helpful.

“Most of the programs trained MULTIPLE (?) domains frequently affected….”

Computerized physical activity was defined as physical activity supervised through a computerized system (line 193). However, it is stated that one of the included articles did not include supervision (line 341), suggesting this study should not have been included. Please clarify.

One study of computerized physical activity programs “did not specify the type of physical activity proposed”. Suggest including a comment on how this physical activity was described in the original paper.

The first paragraph of the discussion comments on previous literature, including “two studies evaluating the efficacy of a combination of computerized cognitive stimulation and physical activity”.  It is unclear what studies are being referred to here, whether these comments refer to the current review or previous studies.

There is some data regarding the effect of mind-body exercise (e.g., yoga, qigong) on cognition. For example, see the recent review by Campbell KL, Phys Ther. 2020 Mar 10;100(3):523-542. doi: 10.1093/ptj/pzz090. While these interventions were excluded in this review, would suggest providing some comment on this as an area of consideration when thinking about potential physical activity interventions.

“Previous reviews are already obsolete because…”. Would suggest that this is not a compelling argument, as many of the findings in this review are consistent with the previous reviews.  Would suggest that the strengths of the current review, in terms of focusing on computerized interventions, can be highlighted without this criticism of previous reviews.

One of the unique aspects of this review is the inclusion of patients with all ages. However, little attention is paid to this. What are the strengths and limitations of this (e.g., for clinicians who work in the pediatric field)?

Author Response

Point 1: Please provide operational definitions for “Cognitive stimulation” and “cognitive training”. There is some description of this in the discussion, but would suggest moving this to the introduction in order to better orient the reader.

Response 1:  As suggested the paragraph concerning the operational definitions of cognitive stimulation, cognitive training and cognitive-behavioral therapy has been moved to the introduction section (85-96). Furthermore, to harmonize the introduction, the definition of physical activity has been included.

Point 2: The PICO elements are not clearly reflective of the search. Would suggest that Intervention = computerized physical activity, computerized cognitive stimulation and combined intervention. “Comparison” should reflect the comparison arms that were of interest, for example usual care, waitlist control, or any other intervention. It seems in this study that any comparison group was accepted. “Outcome” should be “cognitive functioning” or “cognition”.

Response 2: As suggested the PICO elements have been revised. More specifically, the intervention question has been changed as suggested, with the words: “computerized physical activity, computerized cognitive stimulation and combined intervention”. In the comparison question we have written the comparison arms of interest: “usual care, wait-list group or any other intervention other than computerized intervention”.  Finally in the outcome item we have written: “cognitive functioning”.

Point 3: Materials and Methods – It is stated that authors decided not to register the protocol. Suggest to either describe why this decision was made, or just state that “the protocol was not registered”.

Response 3: As suggested in the materials and methods section, it has been written “the protocol was not registered”.

Point 4: Search Strategy item c)  - recommend “assessing their impact on cognition”, rather than “benefit”

Response 4: As suggested the word “benefit” has been changed with “impact”.

Point 5: Throughout the methods, suggest using past tense to describe what was done, not what will be done (e.g., “we also reviewed references…” rather than “we will review references…”)

Response 5: We totally agree with this suggestion, and the past tense has been used in all the method section.

Point 6: For clarity to the reader, I suggest moving the description of included cognitive training and physical activity interventions, to the section describing study eligibility criteria.

Response 6: As suggested the description of included cognitive training and physical activity has been moved to the search strategy section (204-213)

Point 7: Figure 3 – The title contains a typo (“percentage”) but also does not seem to be illustrative for this figure.  Would also suggest re-organizing the figure to align with the order presented in Figure 2 (e.g., Randomisation process in the top row, followed by Deviations from intended interventions, etc.).

Response 7: As the title was suggested to not be illustrative for the figure, it has been deleted. Moreover, items of the figure 3 have been re-organized to follow the same order of items in the figure 2.

Point 8: Characteristics of computerized cognitive stimulation studies, re: Design. It is stated that there are 10 RCTs and six feasibility/pilot studies.  This conflicts with column 1 in Table 1, indicating 12/4. Some clarification to orient the reader is required.

Response 8: We completely agree with this comment, the numbers on table 1 were wrong and have been changed.

Point 9: Characteristics of computerized cognitive stimulation programs. While the program names are listed, providing information on the nature of these programs (i.e., what training elements are included) will better help the reader understand what evidence is available. Linking it back to an operational definition of cognitive stimulation (see comment 1) would be most helpful.

Response 9: As suggested, the operational definition of cognitive stimulation and of cognitive training and cognitive behavioral therapy have been included in the introduction. Doing so, the reader should understand better the nature of the listed program in the section “characteristic of computerized cognitive stimulation program”.

Point 10: “Most of the programs trained MULTIPLE (?) domains frequently affected….

Response 10: The word “multiple” has been added, as suggested

Point 11: Computerized physical activity was defined as physical activity supervised through a computerized system (line 193). However, it is stated that one of the included articles did not include supervision (line 341), suggesting this study should not have been included. Please clarify.

Response 11: The word “supervision” was wrongly used, and has been changed with “delivered”. Indeed, all physical activities delivered through a computerized device were included in the study.

Point 12: One study of computerized physical activity programs “did not specify the type of physical activity proposed”. Suggest including a comment on how this physical activity was described in the original paper.

Response 12: As suggested, more information concerning the study have been included in the section 3.2.2 “characteristics of programs”. 

Point 13: The first paragraph of the discussion comments on previous literature, including “two studies evaluating the efficacy of a combination of computerized cognitive stimulation and physical activity”.  It is unclear what studies are being referred to here, whether these comments refer to the current review or previous studies.

Response 13: We totally agree with this comment, that sentence was confusing and it was deleted.

Point 14: There is some data regarding the effect of mind-body exercise (e.g., yoga, qigong) on cognition. For example, see the recent review by Campbell KL, Phys Ther. 2020 Mar 10;100(3):523-542. doi: 10.1093/ptj/pzz090. While these interventions were excluded in this review, would suggest providing some comment on this as an area of consideration when thinking about potential physical activity interventions.

Response 14: As suggested a comment, basing on the suggested review has been provided in the discussion.

Point 15: “Previous reviews are already obsolete because…”. Would suggest that this is not a compelling argument, as many of the findings in this review are consistent with the previous reviews.  Would suggest that the strengths of the current review, in terms of focusing on computerized interventions, can be highlighted without this criticism of previous reviews.

Response 15: As suggested, that argument has been deleted

Point 16: One of the unique aspects of this review is the inclusion of patients with all ages. However, little attention is paid to this. What are the strengths and limitations of this (e.g., for clinicians who work in the pediatric field)?

Response 16: We totally agree with this comment, and a paragraph concerning the lack of research in the oncology field on elderly and pediatric population has been included in the discussion.

Reviewer 2 Report

The authors were very responsive and improved their manuscript. They have addressed all comments and suggestions on the first version of the article.

Minor comments at this time:

The article needs additional (language) editing.

For example,

p. 4, ln. 148: „overlooked“ (not „overseen“)

p. 7, ln. 273: n = 3538 (not 38)

p. 20, ln. 395: „hormone therapy“

p. 20, ln. 405/423: „assessment after the post-intervention assessment“ = follow-up assessment

p. 21: Table numbering mismatch in caption (Table 3)

Table 4 needs some editing

Figure 3:

„In percentage“ (not „as precentage“); „intention-to-treat“ (not „treat to intention“); to me it is unclear why “intention-to-treat” was added here.

Author Response

Point 1: Minor comments at this time:

The article needs additional (language) editing.

For example,

  1. 4, ln. 148: „overlooked“ (not „overseen“)
  2. 7, ln. 273: n = 3538 (not 38)
  3. 20, ln. 395: „hormone therapy
  4. 20, ln. 405/423: „assessment after the post-intervention assessment“ = follow-up assessment
  5. 21: Table numbering mismatch in caption (Table 3)

Table 4 needs some editing

Figure 3:

„In percentage“ (not „as precentage“); „intention-to-treat“ (not „treat to intention“); to me it is unclear why “intention-to-treat” was added here.

Response 1: All suggestions have been taken in account and used to implement the article.

Reviewer 3 Report

Thank you for revising this important topic of cognitive impairment in our cancer population.  Interventions of computerized cognitive training/stimulation and physical activities were the primary focus of this systematic review.  Your time for these revisions is appreciated as the manuscript is greatly improved.  

A few corrections:

Page 5, line 162: your first research question should contain OR, not AND

Page 5, line 169: make the O in PICO capital, not PICo

Page 6, line 203: restate sentence to reflect past tense; "We reviewed ...."

Page 30, line 16: recommend changing the phrasing from "the last years" to "recent years"

Page 30, line 21: need to say and/or for the interventions in this systematic review

Author Response

Point 1: A few corrections:

Page 5, line 162: your first research question should contain OR, not AND

Page 5, line 169: make the O in PICO capital, not PICo

Page 6, line 203: restate sentence to reflect past tense; "We reviewed ...."

Page 30, line 16: recommend changing the phrasing from "the last years" to "recent years"

Page 30, line 21: need to say and/or for the interventions in this systematic review

Response 1: All suggestions have been taken in account and used to implement the article.